# Synergistic Antifungal Interactions between Antibiotic Amphotericin B and Selected *1*,*3*,*4*-thiadiazole Derivatives, Determined by Microbiological, Cytochemical, and Molecular Spectroscopic Studies

**DOI:** 10.3390/ijms24043430

**Published:** 2023-02-08

**Authors:** Agnieszka Dróżdż, Dominika Kubera, Adrianna Sławińska-Brych, Arkadiusz Matwijczuk, Lidia Ślusarczyk, Grzegorz Czernel, Dariusz Karcz, Alina Olender, Agnieszka Bogut, Daniel Pietrzak, Wojciech Dąbrowski, Andrzej Stepulak, Alicja Wójcik-Załuska, Mariusz Gagoś

**Affiliations:** 1Department of Cell Biology, Maria Curie-Sklodowska University, Akademicka 19, 20-033 Lublin, Poland; 2Department of Biophysics, University of Life Sciences, Akademicka 13, 20-950 Lublin, Poland; 3ECOTECH-COMPLEX—Analytical and Programme Centre for Advanced Environmentally-Friendly Technologies, Maria Curie-Sklodowska University, Głęboka 39, 20-033 Lublin, Poland; 4Department of Chemical Technology and Environmental Analytics, Cracow University of Technology, 31-155 Krakow, Poland; 5Chair and Department of Medical Microbiology, Collegium Universum, Medical University of Lublin, Chodźki 1 Street, 20-093 Lublin, Poland; 6I Clinic of Anaesthesiology and Intensive Therapy with Clinical Paediatric Department, Medical University of Lublin, Jaczewskiego 8, 20-090 Lublin, Poland; 7Department of Biochemistry and Molecular Biology, Medical University of Lublin, 20-093 Lublin, Poland; 8Department of Physical Therapy and Rehabilitation, Medical University of Lublin, 20-093 Lublin, Poland

**Keywords:** amphotericin B, *1*,*3*,*4*-thiadiazole derivatives, synergy mechanisms, FTIR spectroscopy, molecular spectroscopy, molecular aggregation

## Abstract

In recent years, drug-resistant and multidrug-resistant fungal strains have been more frequently isolated in clinical practice. This phenomenon is responsible for difficulties in the treatment of infections. Therefore, the development of new antifungal drugs is an extremely important challenge. Combinations of selected *1*,*3*,*4*-thiadiazole derivatives with amphotericin B showing strong synergic antifungal interactions are promising candidates for such formulas. In the study, microbiological, cytochemical, and molecular spectroscopy methods were used to investigate the antifungal synergy mechanisms associated with the aforementioned combinations. The present results indicate that two derivatives, i.e., C1 and NTBD, demonstrate strong synergistic interactions with AmB against some *Candida* species. The ATR-FTIR analysis showed that yeasts treated with the C1 + AmB and NTBD + AmB compositions, compared with those treated with single compounds, exhibited more pronounced abnormalities in the biomolecular content, suggesting that the main mechanism of the synergistic antifungal activity of the compounds is related to a disturbance in cell wall integrity. The analysis of the electron absorption and fluorescence spectra revealed that the biophysical mechanism underlying the observed synergy is associated with disaggregation of AmB molecules induced by the *1,3,4*-thiadiazole derivatives. Such observations suggest the possibility of the successful application of thiadiazole derivatives combined with AmB in the therapy of fungal infections.

## 1. Introduction

The incidence of fungal infections is steadily increasing every year. Invasive fungal infections are a global problem causing approximately 1.7 million deaths annually [1]. Candidiasis is one of the most common mycoses. *Candida* fungi constitute the natural physiological flora of humans; they are present on the skin and in the gastrointestinal and reproductive systems. However, there are many factors that may lead to their overgrowth and the development of a fungal infection. Transplant recipients receiving immunosuppressive drugs as well as the elderly and hospitalized patients are particularly vulnerable to the same [2]. Although they are ubiquitous, *Candida* fungi are recognized as the leading cause of nosocomial infections. The most common of the many species isolated from clinical materials include *Candida albicans* (37%), *Candida glabrata* (27%), and *Candida parapsilosis* (14%) [2]. What is particularly alarming is the increase in the incidence of infections with non-albicans species, which often exhibit greater resistance to polyenes and azoles [3,4].

Recently, candidiases have become a major threat to hospitalized COVID-19 patients with a severe course of the disease. Many hospitals worldwide have reported an increase in infections caused by *Candida auris*, which poses a serious threat due to the multidrug resistance of the strain, problems with detection of the fungus, and high mortality rates [5,6]. Currently, drugs from the group of polyenes (amphotericin B) and azoles (fluconazole, itraconazole, voriconazole) are most commonly used in the treatment of fungal infections. Amphotericin B (AmB) is the most effective and most frequently used antibiotic. It is a commonly used antifungal agent that shows severe systemic toxicity. The daily infusion dose considered safe is limited to between 0.5 and 1.5 mg/kg/day depending on the pathogen, the patient’s condition, and the severity of the infection. Higher concentrations of the antibiotic can cause many side effects such as nausea, vomiting, cardiovascular disorders, and kidney damage. In vitro studies have shown that amB concentrations ranging from 5 to 10 µg/mL caused abnormal morphology and reduced proliferation of osteoblasts and fibroblasts. Concentrations of 100 µg/mL and above caused cell death [7,8]. AmB is mainly used in severe systemic mycoses of, e.g., lungs and airways, the nervous system, and the gastrointestinal tract. The low number of effective antibiotics, the high cell toxicity of available medications, and the increasing drug resistance of pathogens necessitate the search for new potential drugs and new combination therapies. The development of new antifungal agents is difficult due to the eukaryotic nature of *Candida* cells, which results in a limited number of drug targets [1,2,9,10,11].

Antifungal agents should exhibit selective toxicity towards pathogens while simultaneously remaining safe for humans. Combination therapies are used to improve treatment outcomes and reduce drug toxicity. Major synergistic mechanisms may be induced by increasing membrane permeability, reducing drug efflux from the cell, inhibiting biofilm formation, and inhibiting heat shock protein 90 (Hsp90) synthesis [12,13]. However, some studies have demonstrated antagonism between AmB and azoles. Drugs which inhibit ergosterol biosynthesis, such as fluconazole in combination therapy with AmB, may have an antagonistic effect due to the fact that ergosterol is a target molecule for AmB. This is a big problem in the treatment of mycoses because such drugs induce the resistance of fungal pathogens to AmB itself [14]. Thus, we focused on the search for substances exhibiting synergy with AmB. This study involved a group of synthetic *1*,*3*,*4*-thiadiazole derivatives characterized by antimicrobial activity and synergy with AmB, as reported in our previous publications [15,16,17]. The results presented in our studies to date demonstrated strong antifungal activity of 4- (5-methyl-1,3,4-thiadiazol-2-yl)benzene-1,3-diol (compound C1) against various *Candida* species, with an MIC100 value in the range from 8 to 96 μg/mL and a low toxicity towards animal cells. Moreover, its synergy with AmB was found to reduce the MIC100 value even by tens of times [15,18]. Based on the previous results, we selected a group of *1*,*3*,*4*-thiadiazole derivatives designated as C1, NTBD, Et-NTBD, and C1-NTBD.

The aim of the present study was to analyze the activity of C1, NTBD, Et-NTBD, and C1-NTBD against *Candida* and the compounds’ interaction with AmB. The concentrations of the *1*,*3*,*4*-thiadiazole derivatives and AmB exhibiting synergistic antifungal activity against *Candida* were determined using the microdilution method. Both laboratory strains and clinical isolates of the selected *Candida* yeasts were analyzed. Furthermore, the selectivity index (SI) and the antibiofilm activity of the tested compound were determined. ATR-FTIR spectroscopy was used to analyze the molecular mechanisms of the antifungal effects of the antibiotics and their combinations on *C. albicans* cells. Additionally, the use of electron absorption and fluorescence methods contributed to the identification of a biophysical mechanism that was most likely responsible for the synergistic interaction between the selected *1*,*3*,*4*-thiadiazoles with AmB.

## 2. Results

### 2.1. Antifungal Activity of Compounds

To determine the MIC value of the analyzed *1*,*3*,*4*-thiadiazole derivatives, the broth microdilution method was used in accordance with the standard methodological guidelines recommended by the Clinical and Laboratory Standards Institute (CLSI) [19]. The structural formulas of the tested *1*,*3*,*4*-thiadiazole derivatives are presented in Figure 1.

The activity of all the tested substances against reference strains is shown in the graphs (Figure 2). All substances were tested in the concentration range of 0.5–256 µg/mL. The results of these analyses revealed that, depending on the concentration, some compounds inhibited the growth of *Candida* fungi. In the case of thiadiazoles C1 and C1-NTBD, the values of the minimum concentration inhibiting fungal growth to 100% (MIC100) varied from 64 µg/mL to 128 µg/mL, depending on the strain. The MIC100 value of the Et-NTBD compound in the case of the *C. parapsilosis* strain was 16 µg/mL, whereas *C. albicans* was characterized by a lower sensitivity to this compound, as the highest concentration used caused growth inhibition only by approximately 40%. Thiadiazole NTBD showed no significant inhibitory activity against the reference strains used in the experiment.

### 2.2. Interactions of the Thiadiazoles with the Antibiotic

The checkerboard test was used to assess the interaction of the analyzed compounds with AmB. To obtain the most reliable results, readings were taken after 48-h incubation. The results are presented in Table 1. Some of the compounds used separately at the concentrations selected for the experiment showed no or very low inhibitory activity. Consequently, the MIC value of some compounds was not determined. The preliminary experiment was carried out with the use of two reference strains: *C. albicans* NCPF 3153 and *C. parapsilosis* ATCC 22019, which are characterized by different levels of sensitivity to the tested antibiotics. Thiadiazoles C1 and NTBD applied in the concentration range of 2–64 µg/mL exhibited synergistic (∑FIC ≤ 0.5) interactions with AmB against the two reference strains. No synergistic interactions of the Et-NTBD and C1-NTBD compounds with AmB were observed. There were no antagonistic interactions (∑FIC ≥ 4) at any concentration of the tested compounds applied to the strains.

Two compounds, C1 and NTBD, were selected for tests on the clinical strains, as they demonstrated the best synergistic effect with AmB against the reference strains. The results are shown in Table 2 and Table 3. C1 with the antibiotic exhibited an additive and synergistic effect only against the *C. albicans* and *C. parapsilosis* isolates. The synergistic effect was observed at concentrations ranging from 4 to 32 µg/mL in the case of *C. albicans* and 2–32 µg/mL in the tests on *C. parapsilosis*. The aforementioned doses of C1 resulted in a four to eight-fold decrease in the MIC100 AmB value. At the same time, a contrary effect was registered in the experiments on *C. krusei* and *C. glabrata*, where either no interaction or an antagonistic effect between AmB and C1 was observed at all the concentrations.

The application of the combination of NTBD and AmB on the clinical *Candida* strains yielded different effects compared with those observed in the tests on the reference strains. There was no synergistic or additive effect of NTBD and AmB on the clinical isolates. No interactions were observed at any of the concentrations used in the tests with *C. albicans* and *C. glabrata*. In the case of the other strains, we observed an antagonistic interaction of NTBD concentrated at 2–64 µg/mL when used against *C. krusei* and at 64 µg/mL against *C. parapsilosis*.

### 2.3. Cell Morphology Observations

Morphological changes were observed in reference cells incubated with the thiadiazoles as part of the experiment (Figure 3). Cell aggregates were formed in samples incubated with the addition of C1 and NTBD. The treatment of yeast cells with the combination of C1/NTBD and AmB resulted in the death of most cells in both cases, and the few remaining cells formed clusters. These enlarged cells exhibited altered morphology.

### 2.4. Selectivity Index of the 1,3,4-thiadiazole Derivatives

A selectivity index (SI) was calculated to determine the compound usefulness in the therapy of fungal infections in humans. The SI value higher than one indicates that drug efficacy against fungal cells is greater than the toxicity against host cells [20,21]. Table 4 shows the SI values of tested thiadiazole derivatives (C1 and NTBD) which was calculated using the MIC drug alone and in combination with AmB. As presented in Table 4, C1 alone and in combinations displayed higher SI than NTBD against both *Candida* strains. The potential therapeutic benefit of NTBD against *C. albicans* may be achieved for concentration 16 μg/mL in NTBD + AmB, while against *C. parapsilosis* for the range 4–16 μg/mL in NDBD + AmB.

### 2.5. Antibiofilm Activity of the Tested Compounds

The biofilms of two reference strains—*C. albicans* and *C. parapsilosis—*were exposed to the concentrations of C1/NTBD and AmB that demonstrated synergistic effect in the interaction study presented in Section 2.2. The metabolic activity of the biofilms was analyzed using the MTT assay, where MTT is reduced and converted into a formazan salt which reflects the quantity of metabolically active fungal cells. The results are presented in Figure 4. The amount of metabolically active cells in the 48 h biofilms decreased at all dosages of C1/NTBD + AmB combinations by approximately 29–40% in the case of *C. albicans* and 24–44% in the tests on *C. parapsilosis* compared with the control. Higher concentrations of C1 (32μg/mL and 16 μg/mL) in combination with AmB gave a better biofilm-inhibiting effect than the antibiotic alone at the same concentration in the case of *C. albicans*. Thiadiazole derivatives C1 and NTBD applied independently showed no significant activity against the reference strains used in the experiment.

### 2.6. ATR-FTIR Analysis of Biomolecular Mechanisms of Activity in the Combinations of 1,3,4-thiadiazole Derivatives with AmB in C. albicans

The mean ATR-FTIR spectra of the control *C. albicans* cells and *C. albicans* treated with C1 (8 µg/mL), AmB (0.03 µg/mL), and their combination, respectively, are presented in Figure 5A–C, while the mean IR spectra of isolates exposed to NTBD (4 µg/mL), AmB (0.06 µg/mL), their combination, and the corresponding control are shown in Figure 6A–C. The concentrations of the thiadiazoles and AmB were adjusted so that the amount of inoculum grown was sufficient for the FTIR measurements and, at the same time, the inhibition of fungal growth was observed. In order to enhance spectral differences and resolve the problem of overlapping components in IR absorption bands, reversed second derivatives of the aforementioned spectra, were calculated and presented in Figure 5a–c and Figure 6a–c, respectively. Based on the reversed second derivatives, the spectral assignments were performed as presented in Table 5.

The differences in the intensity of bands identified for *C. albicans* treated with C1, AmB, and their combination, as well as NTBD, AmB, and their combination, respectively, compared with the control, were analyzed based on the integral areas of peaks calculated for the reversed second derivatives of the IR spectra. The statistical significance of the observed changes was assessed with the Mann–Whitney U test at the significance level of 5%. Additionally, statistical trends were analyzed at the significance level of 10%. The results of the statistical analysis conducted for selected absorption bands are presented in Figure 7 and Figure 8.

**Table 5 ijms-24-03430-t005:** Identification of ATR-FTIR bands in *C. albicans* spectra; str.—stretching, asym.—asymmetric, sym.—symmetric, def.—deformation, bend.—bending vibratiosns [18,22,23,24,25,26,27,28,29,30,31].

Band [cm^−1^]	Origin	Characteristic
2958	CH_3_ str. asym.	Lipid and polysaccharide level
2918	CH_2_ str. asym.	
2870	CH_3_ str. sym.	
2849	CH_2_ str. sym.	
1653	C=O str., NH bend. (Amide I)	Protein and chitin level
1543	NH bend., C-N str. (Amide II)	Protein and chitin level
1455	CH_2,_ CH_3_ def.	Lipid, protein, and polysaccharide level
1369		
1252	NH def. (Amide III)	Protein and chitin level
1237	P=O str.	Phosphomannan level
1200	PO_2_ str. asym.	Phosphomannan level
1153	β(1→3) glucans	β(1→3) glucan level
1105	Glycogen, β(1→3) glucans	Glycogen and β(1→3) glucan level
1078	Glycogen	Glycogen level
1045	Glycogen, mannans	Glycogen and mannan level
1022	Glycogen	Glycogen level
992	β(1→6) glucans	β(1→6) glucan level
965	Mannans	Mannan level
933	Glucans, mannans	Glucan and mannan level
888		

#### 2.6.1. C1, AmB, and C1 + AmB Effect

As shown in Figure 7, the ATR-FTIR analysis revealed a statistically significant decrease in the intensities of 2958 cm^−1^ and 2918 cm^−1^ absorption bands originating from CH_3_ and CH_2_ asymmetric stretching vibrations in *C. albicans* treated with AmB at a concentration of 0.03 µg/mL, compared with the control. In turn, the cells exposed to C1 exhibited a decrease in the intensity of the 2958 cm^−1^ band, while the level of 2918 cm^−1^ increased. Additionally, as shown in Figure 5a, the maxima of both bands were slightly shifted towards higher wavenumbers relative to their position in the control group. In the C1-treated *C. albicans*, the elevated intensity of the 1455 cm^−1^ band assigned to CH_2_ and CH_3_ deformation vibrations was also observed. Cells treated with C1 + AmB did not exhibit any statistically significant differences in terms of the intensity of the 2958 cm^−1^ and 2918 cm^−1^ bands; however, a shift of the 2958 cm^−1^ band was observed, compared with the control (Figure 5a). Furthermore, a significant increase in the 1455 cm^−1^ band was noticed in this group, relative to the control.

In the AmB-treated *C. albicans*, a decrease in the Amide I (1653 cm^−1^) level and an increase in the Amide III (1252 cm^−1^) level were observed. The C1-treated cells were characterized by a statistically significant increase in the Amide I and Amide III intensities, while reduced content was noticed for the Amide II (1543 cm^−1^) band. The cells treated with the C1 + AmB combination showed a lower level of Amides I and II, compared with the control, and increased intensity of Amide III, as in the case of *Candida* exposed to C1 and AmB.

The levels of both β(1→3) and β(1→6) glucans, with absorption at 1153 cm^−1^ and 992 cm^−1^, respectively, were significantly lower in the C1-, AmB-, and C1 + AmB-treated *C. albicans* than in the control group. The decreases were the most pronounced for cells treated with the C1 + AmB combination. Additionally, in the case of *Candida* exposed to C1 and C1 + AmB, shifts of β(1→3) glucan band maxima towards higher frequencies were observed, compared with the control (Figure 5c). Interestingly, the intensity of the 888 cm^−1^ absorption band related to the content of total glucans and mannans was significantly elevated in the C1-, AmB-, and C1 + AmB-treated cells, compared with the control.

The intensities of the 965 cm^−1^ absorption band assigned to mannans were elevated in *C. albicans* exposed to C1, AmB, and C1 + AmB, compared with the control. Similarly, in the three groups, an increase in the 1231 cm^−1^ band originating from the P=O bond stretching vibration in the phosphate groups was observed. In turn, the level of the 1200 cm^−1^ band assigned to PO_2_^−^ asymmetric stretching vibrations was significantly higher in the C1- and C1 + AmB-treated cells, relative to the control.

The level of the 1045 cm^−1^ absorption band related to the content of mannans and glycogen was significantly elevated in *C. albicans* exposed to AmB, while its level in the C1- and C1 + AmB-treated cells was lowered, compared with the control group. The intensities of the 1078 cm^−1^ and 1022 cm^−1^ peaks assigned to the glycogen content decreased in the case of cells treated with C1 and C1 + AmB, but only the former peak was significantly lower relative to the control in the AmB treatment of *Candida*. Additionally, the 1078 cm^−1^, 1045 cm^−1^, and 1022 cm^−1^ band maxima centered in the controls in *Candida* treated with C1 and C1 + AmB were shifted towards lower wavenumbers.

**Figure 7 ijms-24-03430-f007:**
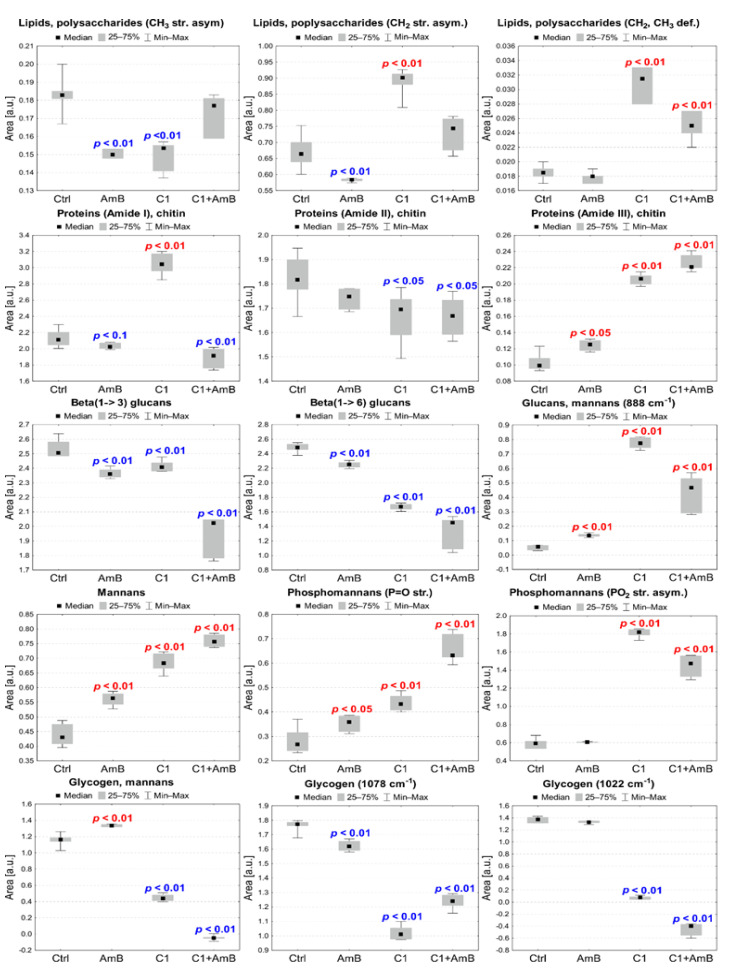
The median, minimal, and maximal values of intensities of selected absorption bands presenting differences between *C. albicans* exposed to AmB, C1, and C1 + AmB treatment and control (Ctrl) group. The *p*-values of Mann–Whitney U test for statistically significant changes compared with the control are placed on the charts (red—ncreases, blue—decreases).

#### 2.6.2. AmB, NTBD, and NTBD + AmB Effect

Biochemical anomalies, compared with the control, were also detected for *C. albicans* exposed to AmB (0.06 µg/mL), NTBD, and the NTBD + AmB combination, as presented in Figure 8. Statistically significant increases in the intensities of the 2958 cm^−1^ and 2918 cm^−1^ absorption bands, originating from CH_3_ and CH_2_ asymmetric stretching vibrations, were observed in the cells treated with AmB at a concentration of 0.06 µg/mL, compared with the control group. The NTBD-treated *C. albicans* showed a relevant elevation of the 2958 cm^−1^ band content. In turn, in the case of cells exposed to the NTBD + AmB combination, the most pronounced increases were observed in both 2958 cm^−1^ and 2918 cm^−1^ and in the 1455 cm^−1^ band assigned to CH_2_ and CH_3_ deformation vibrations. The analysis of Figure 6a did not reveal any shifts in the position of the maxima of the aforementioned bands, compared with the control.

The Amide I (1653 cm^−1^), Amide II (1543 cm^−1^), and Amide III (1252 cm^−1^) intensities were significantly elevated in *C. albicans* treated with AmB, NTBD, and their combination, compared with the control. The most pronounced changes were detected in the case of the cells incubated with NTBD + AmB.

The content of β(1→3) glucans (1553 cm^−1^) and β(1→6) glucans (992 cm^−1^) decreased significantly in the cells exposed to AmB and NTBD + AmB, while a lower level of only β(1→6) glucans in relation to the control was observed in the case of the NTBD-treated *Candida*. Additionally, in *C. albicans* treated with the NTBD + AmB combination, shifts in the β(1→3) and β(1→6) glucan band maxima towards higher frequencies were observed (Figure 6c). The intensity of the 888 cm^−1^ absorption band related to the total content of glucans and mannans was significantly elevated in the AmB, NTBD, and NTBD + AmB treatments, compared with the control.

A significant increase in the mannan absorption band (965 cm^−1^) and the P=O bond stretching vibration in the phosphate groups (1231 cm^−1^) was detected for *C. albicans* treated with AmB, NTBD, and their combination, compared with the control. In turn, the level of the 1200 cm^−1^ band of PO_2_^−^ asymmetric stretching vibrations was significantly higher than in the control in cells exposed to AmB and NTBD + AmB.

The intensity of the 1045 cm^−1^ absorption band assigned to total glycogen and mannan content was significantly diminished in C*. albicans* treated with AmB, NTBD, and their combination in relation to the control. Furthermore, in the case of cells exposed to AmB and NTBD + AmB, a decrease in the content of glycogen, with an absorption at 1078 cm^−1^ and 1022 cm^−1^, was observed.

**Figure 8 ijms-24-03430-f008:**
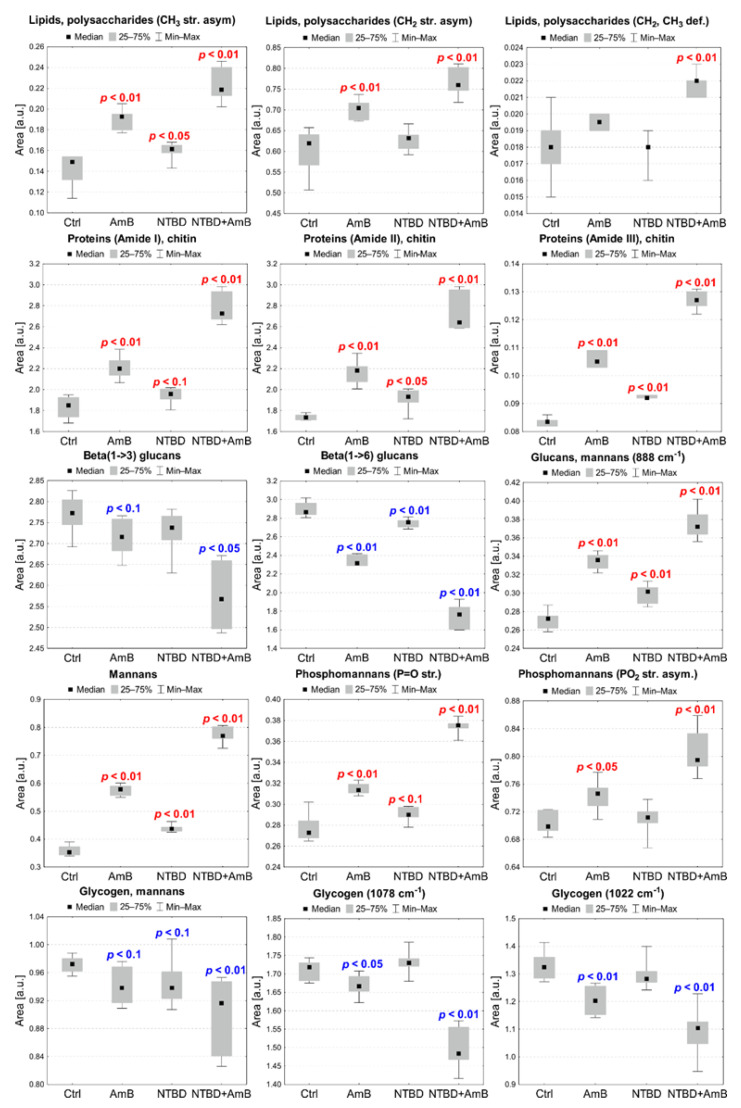
Median, minimal, and maximal values of intensities of selected absorption bands showing differences between *C. albicans* exposed to the AmB, NTBD, and NTBD + AmB treatments and the control (Ctrl) group. The *p*-values of the Mann–Whitney U test for statistically significant changes compared with the control are placed on the charts (red—increases, blue—decreases).

### 2.7. Biophysical Mechanism of Action of Systems with 1,3,4-thiadiazole Derivatives and AmB

The extremely interesting biological properties of C1 and NTBD molecules that show synergy in combination with AmB encouraged an attempt to provide a preliminary elucidation of the biophysical mechanism of these systems in the next step of the research. To this end, absorption and fluorescence spectra were determined. Figure 9 shows the absorption spectra of C1, AmB, and the C1+ AmB system exhibiting synergic effects (Figure 9A) and the spectra of NTBD, AmB, and the NTBD + AmB systems showing synergy in the PBS buffer. To visualize the effect, each spectrum, especially that of the synergistic combination, was measured at several concentrations of the tested compounds. For the sake of clarity of presentation, only three concentrations evidencing the exact effect are shown. As can be seen in Figure 9A, C1 has a characteristic spectrum with a wide maximum at 320 nm, which is characteristic of the π → π * electron transition in this chromophore system [32]. This spectrum is underpinned on the long-wavelength side by a band with a maximum of ~350 nm. As known from the excitonic fission theory and our previous detailed studies, this band is characteristic of aggregated forms of the compound that may be present in the buffer system [33,34]. The same panel shows the spectrum of AmB. As can be seen, AmB has a characteristic band with a maximum at 340 nm. In the literature, this band is mainly associated with aggregated forms of the AmB molecule or with the potential presence of oxidized forms of the molecule [35,36]. It seems that the dominant role in the present case is played by the aggregated AmB forms. However, the measurement of the absorption spectra of the synergistic systems shows a shift of the band with the maximum at 340 nm to 330 nm and a clear broadening of the band. This band seems to be a sum of spectra characteristic of the C1 and AmB molecules. Similar effects are shown in Figure 9B presenting the electron absorption spectra for NTBD, AmB, and NTBD + AmB.

However, since NTBD in PBS has a band with a maximum at 340 nm (π → π * electron transition), the band of the aggregated form has a maximum at around 350 nm (less clear than in the case of C1), which is rather a long-wavelength shift of the band [33,37]. The AmB spectra are obviously analogous to those shown in Figure 9A. In turn, in the case of the synergistic NTBD + AmB system, there are spectra characterized by high absorbance with a maximum at 335 nm. This band gains in intensity, especially with the increase in the concentration of the compounds, and this is most probably a sum of the absorption spectra of NTBD and AmB, i.e., an effect similar to that shown in Figure 9A for the C1-containing system. However, the spectra of C1 and AmB were more clearly separated in the case of C1, whereas the bands of the aggregated forms in the system with the compound NTBD overlapped to a greater extent. Additionally, it is worth emphasizing that the aggregated forms of the *1*,*3*,*4*-thiadiazole compounds visible in the absorption spectra also differ significantly. There is a clear band in the case of C1, while the band of the aggregated form of the NTBD molecule quite clearly enters the monomer band, thereby causing its evident shift. This can also be clearly seen in the synchronous spectra, where the bands of the aggregated forms of molecules NTBD alone are definitely more intense.

Panels C and D in Figure 9 show fluorescence emission spectra obtained at the maximum excitation of the bands characteristic of the synergistic systems corresponding to the electron absorption spectra shown in Figure 9A,B. As can be seen in Figure 9C, for excitation at 350 nm, there is a characteristic emission spectrum of AmB with one of the maxima at 485 nm [35,36]. In turn, the appearance of a very intense fluorescence emission band with a maximum at ~440 nm can be observed for the same excitation in the synergistic system. The intensity of this band is slightly reduced in the C1 + AmB system. This is probably a result of specific interactions between C1 and AmB, e.g., the association of molecules or energy transfer between the chromophores. This phenomenon requires further investigation and will be the focus of our future research. A very similar effect to that shown in Figure 9C can be observed for the NTBD compound in the system characterized by synergistic properties presented in Figure 9D. In this case, the increase in the concentration of NTBD in the synergistic system with AmB is accompanied by the appearance of an emission maximum at around 500 nm characteristic for such molecules in aqueous solutions [38]. There is also a fairly characteristic dual fluorescence effect with a maximum at about 430 and 500 nm, which is quite surprising for this system, given the presence of AmB in aqueous solutions and the phenomenon of excited state intramolecular proton transfer (ESIPT). Clearly, the emission comes mainly from the NTBD molecule; in Figure 9C, in the C1 + AmB combination, there is practically no fluorescence originating from the AmB molecule after this excitation. It is evident in this case that the interaction between NTBD and AmB must cause quite a considerable change in the substrate hydrophobicity. This facilitates NTBD interactions and enhances the ESIPT process.

As a supplement to the spectroscopic part of the study, Figure 10A,B show the resonance light scattering (RLS) spectra [39]. The spectra seem to be extremely important not only for this study but also for our future research on this topic. Figure 10A,B also shows the RLS spectra recorded for C1 in PBS and NTBD in PBS. The AmB molecule has fairly intensive RLS bands in PBS due to its predilection for aggregation [35,36]. However, in both the C1- and NTBD-containing synergistic systems, the increase in the concentration of C1 and NTBD is accompanied by an evident decrease in the intensity of the RLS spectra, which indicates the clear disaggregation of AmB in the presence of these compounds. Nevertheless, it should be emphasized that these effects differ noticeably in the presence of C1 molecules compared with those observed in the presence of NTBD, whose aggregation propensity is much higher even when taken separately (Panel B in Figure 10). Notably, the intensity of the RLS spectra in the synergistic system decreases in comparison with the RLS spectra of AmB alone. C1 molecules produce practically no RLS spectra. This finding differs from the result shown in Panel B in Figure 10, as the NTBD molecule itself produces very intense RLS spectra. However, the RLS spectra of the synergistic system, i.e., NTBD + AmB, are characterized by a clear two-fold loss of intensity. Most probably due to their size, C1 molecules interact more intensively with the part of the AmB molecule responsible for the aggregation properties of such systems.

## 3. Discussion

The resistance of pathogenic fungi to AmB is still rarely reported in comparison with their resistance to other antifungal agents. Despite its high efficiency, there are limitations to the usability of this compound due to the undesirable side effects that may emerge during therapy. Substances showing specific synergistic activity are sought out by researchers to reduce the toxicity of commonly used antibiotics [40,41,42]. In the present study, we documented the beneficial synergistic effects of C1 and NTBD combined with AmB as part of a treatment against *Candida* reference strains, while the studied components alone did not engage in such an interaction. In the case of C1, a similar effect was observed in the experiments carried out on clinical *C. albicans* and *C. parapsilosis* isolates. The synergistic activity of C1 allowed a several-fold reduction of the antibiotic dose required for 100% inhibition of fungal growth. This confirmed our previous experiments, which demonstrated the synergistic viability of thiadiazoles against many clinical strains with enhanced resistance to azoles [15]. Nevertheless, the present study showed that the type of interaction may differ depending on strain-specific traits. The C1 compound did not exhibit interactions with AmB when used on *C. glabrata*, while a downright antagonistic effect was observed in case of *C. krusei*. The synergistic effect of NTBD and AmB observed for the reference strains was not present in the context of the selected clinical strains. The combinations of C1 or NTBD with AmB may have therapeutic potential and inhibit the growth of some *Candida* strains. The substantial reduction in the dose of the antibiotic may limit its side effects. However, given the different types of interactions, research should be continued to test a greater number of clinical isolates. Our previous studies suggest that C1 impacts the biogenesis of fungal cell walls by disrupting its integrity [15,18]. The representative morphological images of *C. albicans* cells treated with the AmB and a combination of C1/NTBD for 24 h confirmed the synergy between the two compounds. In combination with AmB, both NTBD and C1 induce the formation of giant cells and emergence of morphological changes, which may lead to premature cell death. The SI analysis revealed that both C1 and NTBD compounds can be cytotoxic for the mammalian cell lines. Especially, the NTBD has shown a SI of 0.07–4.71 which demonstrated selective toxicity for host cells compared with the pathogen. However, for some concentrations of C1 the SI >10 was observed, which suggests that it is promising candidate for in vivo studies [43,44]. Analysis of the thiadiazole derivatives and AmB antibiofilm activity indicate that the C1 + AmB combination is more effective against *C. albicans* than the antibiotic alone. In the case of *C. albicans* biofilms exposed to NTBD + AmB, no relevant differences were observed compared with the AmB treated ones, which may suggest that a higher dose of NTBD in the combination with AmB may be required to enhance the antibiotic action. Similar conclusions can be drawn the in case of C1 + AmB and NTBD + AmB antibiofilm activity against *C. parapsilosis*.

ATR-FTIR spectroscopy was used to explore the influence of the selected *1*,*3*,*4*-thiadiazole derivatives, AmB, and their combinations on the biochemical content of *Candida*, which may shed light on the biomolecular mechanisms underlying the synergistic antifungal interactions of the analyzed compounds and compositions. Regularities in terms of changes in the 1200–800 cm^−1^ IR polysaccharide absorption region, compared with the control, were noticed in the *C. albicans* treatments with selected thiadiazole derivatives, AmB, and their combinations. Polysaccharides are the main building material of the fungal cell wall, whose inner part is formed by chitin molecules linked with a β(1→3) and β(1→6) glucan core, and the outer layer is composed of mannoproteins [45,46]. β(1→6) glucans play a key role in cell wall stabilization as they link other cell wall components [45,47]. The chitin layer is a skeleton facilitating β(1→3) and β(1→6) glucan crosslinking, which determines cell wall stability and rigidity [45,48,49]. The ATR-FTIR analysis revealed a significant decrease in the β(1→3) and β(1→6) glucan abundance in *Candida* treated with AmB, *1*,*3*,*4*-thiadiazole derivatives, and their combinations; however, the loss in these biomolecules was the most pronounced for the C1 + AmB and NTBD + AmB compositions. Furthermore, spectral shifts of the β(1→3) and β(1→6) glucan maxima towards higher wavenumbers were noticed in both compositions, which may suggest weakened interactions [18]. Compared with the control, the spectra of C1 and C1 + AmB exhibited shifts in the positions of the maxima of CH_3_ and CH_2_ asymmetric stretching vibration absorption bands related to yeasts polysaccharides. Such abnormalities in the IR profile of β-glucans may indicate that the key synergistic antifungal mechanisms of the AmB compositions with the *1*,*3*,*4*-thiadiazole derivatives entail a disruption of cell wall integrity. The β(1→3) and β(1→6) glucan levels were found to correlate with the glycogen content. Glycogen, i.e., an intracellular energy storage carbohydrate in yeasts, has been reported to be covalently linked to the β(1→3) and β(1→6) glucans of the cell wall via the β(1→6)-linked side chain [50,51]. Thus, the observed changes in the level of this biomolecule may be associated, on the one hand, with changes in cell wall integrity arising from the disruption of β-glucan availability, and on the other, with stress conditions related to the consumption of the energy pool [52]. The analysis also showed an increase in the mannan content after the treatment with the analyzed compounds and combinations, but again, this effect was the most pronounced in the case of the C1 + AmB and NTBD + AmB compositions. The mannan level seems to correlate positively with P = O stretching vibrations of phosphate groups, which may suggest elevation of the phosphomannan moiety in cell walls [53,54]. Additionally, there was a positive correlation with the protein-related amide III level in the variant with AmB (0.03 µg/mL), C1, and their composition and a similar correlation between AmB (0.06 µg/mL), NTBD and their combination, and the absorption bands of all amides This may indicate that the observed changes result from an increase in mannoproteins modified by phosphomannosylation in the *Candida* cell wall [55]. Interestingly, correlations were also observed between the level of phosphomannans and CH_2_, CH_3_ deformation as well as asymmetric stretching vibrations in *Candida* exposed to AmB (0.06 µg/mL), NTBD, and NTBD + AmB. It is worth emphasizing that the strongest increase in the mannosyl-phosphate moiety content was also detected in the compositions of AmB and *1*,*3*,*4*-thiadiazole derivatives. Yeast mannans are important for tissue adhesion, virulence, and cell wall integrity [53,56]. The outer layer of mannoproteins is believed to determine the porosity of the yeast cell wall [57,58]. In turn, changes in phosphomannan content have been related to the regulation of the stress response to drought, high osmolarity, or nutrient limitations [59]. Thus, it cannot be unequivocally stated whether the observed elevations in mannan and phosphomannan levels are a result of the interaction between the investigated antimicrobial formulations and these cell wall components are part of *Candida* response to stress conditions. However, it has been reported that the presence of phosphomannans in the *Candida* cell wall determines fungal sensitivity to some antimicrobial compounds [58,59]. This observation together with the results of our current and previous studies may support the assumption that *1*,*3*,*4*-thiadiazoles increase cell sensitivity to AmB by increasing the phosphomannan level in the cell wall [15]. Notably, cell wall mannans and phosphomannans are key for the recognition of pathogens by the host immune system and the initiation of a protective immune response [53]. Therefore, the observed increase in the phosphomannan moiety level after the treatment with the *1*,*3*,*4*-thiadiazole combinations with AmB may give hope for achieving even more effective antifungal activity of these formulations in vivo.

Given the results of the analysis of absorption and electron fluorescence spectra recorded for thiadiazoles and their combinations with AmB, it is possible to propose a biophysical mechanism of the interaction between C1/NTBD compounds and the AmB molecule. Most probably, the addition of C1 or NTBD to an AmB-containing solution causes disaggregation of the antibiotic, as in the case of, e.g., sodium deoxycholate used in hospital treatment. The absorption spectra show the appearance of a characteristic band, which is the sum of the absorption spectra of the individual components. In turn, after AmB disaggregation, the emission spectra mainly show either a characteristic emission from C1 with a maximum at 440 nm or, in the case of NTBD, a strong emission with a maximum at ~500 nm, most likely originating from the *keto* tautomer of this *1*,*3*,*4*-thiadiazole analogue [38]. The AmB molecule monomer exhibits very low fluorescence emissions, especially for this excitation wavelength [60,61]. The most important conclusion, however, can be formulated based on the resonance light scattering RLS spectra, where a significant decrease in the band intensity with an increased dose of the compound (C1 or NTBD) in the buffer can be observed in the synergistic systems. The disaggregation of AmB molecules enhances its antifungal activity and reduces its toxicity substantially, as shown by the literature data [62,63,64,65]. The reduction of AmB toxicity appears to be a very promising effect of the interactions in these combinations. Nevertheless, it should be emphasized that the effects related to the disaggregation of AmB seem to be much stronger in the case of the synergistic system with the C1 molecule, which by itself does not exhibit a strong predilection for aggregation; this may confirm the results of the microbiological tests. The intensity of the RLS spectra of the combination of NTBD with AmB exhibits a more marked decrease; however, as shown in the study, the NTBD molecule has a much greater tendency to aggregate and its RLS spectra are characterized by a much greater intensity than the spectra of the AmB molecule.

## 4. Materials and Methods

### 4.1. Strains

Two reference strains: *C. albicans* NCPF 3153 and *C. parapsilosis* ATCC 22019 as well as *C. albicans*, *C. parapsilosis*, *C. krusei*, and *C. glabrata* clinical isolates were used in the study. The experiments on the reference strains were conducted in the microbiological laboratory of the Department of Cell Biology, Maria Curie-Skłodowska University in Lublin, whereas the clinical strains were analyzed in the Laboratory of the Chair and Department of Medical Microbiology, Medical University of Lublin. The fungi were stored at -70 °C in VIABANK cryovials (BioMaxima). To obtain logarithmic fungal cultures before each experiment, the strains were grown in shaken pre-culture in YPD liquid medium (1% yeast extract, 2% bactopeptone, 2% glucose) in 0.01 mol/L phosphate buffer (PBS), pH 7.0, at 35 °C for 24 h.

### 4.2. Antifungal Substances

Powdered AmB from *Streptomyces* sp. was purchased from Sigma-Aldrich, St. Louis, MI, USA (cat. no. A4888). The purity of the antibiotic powder (HPLC) was approx. 80%, which was taken into account in the calculations of the concentrations. Then, 1 mg of AmB was dissolved in 100 µL of DMSO, supplemented with culture medium to a concentration of 1 mg/mL, and diluted with the culture medium immediately to obtain a stock solution with a concentration of 10 µg/mL. The stock solution of AmB was prepared immediately before each experiment to prevent loss of activity. The stock solutions of the tested compounds were prepared by dissolving 1 mg of the powder in 100 µL of DMSO and diluting the solution with the medium to a concentration of 1 mg/mL. The structural formulas of the tested *1*,*3*,*4*-thiadiazole derivatives are presented in Figure 1 in Section 2.1.

### 4.3. Antifungal Activity of the Tested Compounds

The antifungal activity of the selected antibiotics and thiadiazoles was assessed by determining the minimum inhibitory concentrations (MIC) with the broth microdilution method. Standard methodological procedures recommended by the Clinical and Laboratory Standards Institute (CLSI) were used (the M27-A342 standard for yeast susceptibility testing) [19]. The fungal inoculum for yeasts with a final density of 0.5 × 10^3^–2.5 × 10^3^ cells/mL was prepared in the culture medium. RPMI 1640 medium without phenol red and sodium bicarbonate (Sigma-Aldrich, St. Louis, MI, USA, R8755) buffered to pH 7.0 with 0.165 mol/L of 3-(N-morpholino) propanesulfonic acid (MOPS) and supplemented with 2% glucose to optimize the fungal growth conditions was used in the experiment. The same medium was used in all the fungal cultivation experiments. The antibiotic and thiadiazole stock solutions were used to prepare the final dilutions in broth medium in 96-well microtiter plates. The final DMSO concentration did not exceed 0.5% of the culture medium, and the fungal growth was controlled in a solvent-supplemented medium. After inoculation, the plates were incubated at 35 °C for 48 h. Optical density was determined spectrophotometrically at the wavelength of 600-nm (OD 600) using an EPOCH 2 microplate reader. The complete inhibition of fungal growth after the 48-h incubation was adopted as the MIC100 value. The MIC value was determined by calculating the percentage growth vs. the control in each replicate using the following formula: (OD600 of treated sample)/(OD600 of untreated sample) × 100.

### 4.4. Interactions between Thiadiazoles and Antibiotics

The interactions between the thiadiazoles and the antibiotics against the *Candida* strains were investigated using the checkerboard microdilution technique described elsewhere [15]. Each dilution (50 µL) of each agent was placed in microtiter plate wells to obtain 64 drug/compound combinations. The final drug concentrations after the addition of 100 μL of the inoculum ranged from 0.5 to 64 μg/mL of the thiadiazoles and from 0.016 to 4 μg/mL of amphotericin B, and the final inoculum size was 2.5 × 10^3^ cfu/mL. The plates were incubated at 35 °C for 48 h. The MIC values were read visually after 48 h. The interpretation of the interaction between the two components of the composition with the concentrations used was based on the ∑FIC coefficient, which is the sum of partial concentrations inhibiting the growth of the pathogens. Standard methodological criteria indicating synergy between the respective components against *C. albicans* and *C. parapsilosis* pathogens were adopted. The FIC index was calculated using the following formula: ∑FIC = FIC A + FIC B; where, FIC A is the MIC of drug A in the combination/MIC of drug A alone, and FIC B is the MIC of drug B in the combination/MIC of drug B alone. The following interaction criteria were adopted: ∑FIC ≤ 0.5—synergy, ∑FIC ≥0.5 to 4 indifferent and ∑FIC > 4—antagonism [66].

### 4.5. Morphological Observations of Fungal Cells

Chitin staining with Calcofluore White (Sigma-Aldrich, St. Louis, MI, USA, 18909) was used to visualize the morphology of fungal cells treated with the selected thiadiazole derivatives. *C. albicans* NCPF 3153 cells were transferred to 96-well plates in accordance with the applicable procedure to determine interactions between thiadiazoles and antibiotics. After 24-h incubation, the cells were placed in a solution containing 1 g/L of Calcofluor white and 0.5 g/L of Evans Blue for 5 min; next, the cells were centrifuged and washed with phosphate-buffered saline (PBS) buffer, pH 7.4. The cells were observed under a Nikon Labophot 2 fluorescence microscope at 380–420 nm. The micrographs were taken with a Canon Power Shot A 640 camera.

### 4.6. Selectivity Index Calculation

Half-maximal inhibitory concentration (IC50; concentration of compounds resulting in 50% inhibition of metabolic activity) calculations for the thiadiazoles tested were determined based on the results of the mitochondrial dehydrogenase activity assay (24 h/48 h, MTT) obtained for normal RPTEC cells (published data [67]). IC50 values were calculated by plotting a non-linear regression curve between Log dose and percentage (%) of cell viability (Figure 11) using the GraphPad Prism 5 software (Trail, GraphPad Software, La Jolla, CA, USA). The selectivity index (SI) of tested compounds was calculated using the following formula: SI = IC50 in μg/mL (normal RPTEC cells)/MIC in μg/mL [20]. As NTBD antifungal activity was not observed at 128 μg/mL, a value of 256 μg/mL was used to calculate SI. SI value > 1.0 indicates a drug with efficacy against fungal cells greater than the toxicity against host cells [20,21]. A high selectivity index means that the drugs may be useful in treating fungal infections in humans.

### 4.7. Antibiofilm Activity of the Tested 1,3,4-thiadiazole Derivatives

For *C. albicans* (NCPF 3153) and *C. parapsilosis* (ATCC 22019) biofilm formation, 96-well plates were inoculated with Candida suspensions (1 × 107 cells/mL) and incubated at 35 °C for 48 h. After incubation biofilms were washed twice with PBS and then treated with 200 µL of AmB, C1/NTBD and combination of AmB with C1/NTBD in concentrations which demonstrated synergic interaction against reference strains. The MTT reduction assay was carried out to evaluate cell metabolic activity. Candida biofilms were incubated at 35 °C with 100 µL of 0.5% 3-(4,5-dimethyl-2-thiazolyl)-2,5-diphenyl-2H-tetrazolium bromide (MTT) for 4 h. The liquid was removed from each well, 100 µL of dimethyl sulfoxide (DMSO) was added to solubilize the formed formazan salt. The absorbance was measured at 570 nm using an EPOCH 2 microplate reader. Metabolic activity of biofilm value was determined by calculating of the percentage activity vs. the control in each replicate using the following formula: (absorbance of treated sample)/(absorbance of untreated sample) × 100 [68].

### 4.8. ATR-FTIR Spectroscopy

Attenuated total reflection Fourier transformed infrared spectroscopy (ATR-FTIR) was employed to study the molecular mechanisms underlying the synergistic antibiotic action of thiadiazoles (C1 and NTBD) combined with AmB against *C. albicans*. For this purpose, *C. albicans* isolates were exposed to 48-h incubation with C1 (8 µg/mL), AmB (0.03 µg/mL) and their mixture as well as with NTBD (4 µg/mL), AmB (0.06 µg/mL), and their mixture, as described in Section 4.4. A *C. albicans* isolate incubated for 48 h was used as naive control. The incubation was performed in six replications for each case. Subsequently, the isolates were harvested and centrifuged, the medium was removed, and the samples were fixed with 2.5% paraformaldehyde, which was followed by washing three times with PBS and twice with deionized water.

For ATR-FTIR measurements, 60 ul of the final suspension of each sample was distributed on a ZnSe crystal and air-dried. ATR-FTIR spectra within the wavelength range of 4000–700 cm^−1^ were recorded for the all analyzed isolates using an FTIR VERTEX 70 spectrometer (Bruker Optic GmbH, Germany) with an MCT detector. The IR spectra were sampled at the spectral resolution of 2 cm^−1^, and 64 scans were averaged per each sample spectrum. Calculations for the baseline correction, vector normalization, reverse second derivatives, and the band area were performed using OPUS 7.5 software (Bruker Optic GmbH, Ettlingen, Germany). The Origin Pro 2020b (OriginLab Corporation) was used for graphical processing of the IR spectra and their reverse second derivatives. The statistical analysis of changes in the IR spectra of C1, NTBD, and AmB as well as C1 + AmB and NTBD + AmB treated *C. albicans*, relative to the naive control, was performed with STATISTICA 7.1 software (StatSoft. Inc., Tulsa, OK, USA, 2005). The non-parametric Mann–Whitney U test was applied to assess the statistical significance of the detected differences. The choice of the non-parametric test was dictated by the fact that the analyzed data did not fulfill the assumptions of normality and homoscedasticity, which are necessary for the use of its parametric alternative.

### 4.9. Measurements of Absorption and Electron Fluorescence Spectra

Stock solutions of the compounds (at 1 mg/mL DMSO) were prepared: C1 with a concentration of 4.80 mM, NTBD with a concentration of 2.88 mM, and AmB with a concentration of 1.08 mM. The measurements of absorption spectra, fluorescence emission, and synchronous RLS spectra were carried out at varying concentrations.

The measurements were carried out in a buffered saline PBS solution prepared by dissolving one PBS tablet (Sigma-Aldrich^®^, St. Louis, MI, USA, P4417) in 200 mL of double distilled water, which yielded 0.01 M phosphate buffer containing potassium chloride (0.0027 M) and sodium chloride (0.137 M), pH 7.4, at 25 °C.

The electron absorption spectra were measured for C1 and NTBD using a Bio Varian UV-Vis Cary 300 Dual Beam Spectrophotometer (Middelburg, The Netherlands) equipped with a thermostatted cuvette holder with a 6 × 6 multi-chamber Peltier block. During the measurement, the temperature was controlled by a thermocouple probe (Cary Series II from Varian) placed directly in the sample.

A Cary Eclipse spectrofluorometer (Varian) was used to record the fluorescence emission spectra and the synchronous RLS (Resonance Light Scattering) spectra. All fluorescence spectra were measured at a resolution of 0.5 nm taking into account the spectral characteristics of the lamp and the photomultiplier. Resonance light scattering (RLS) was measured by synchronous scanning of the excitation and emission monochromator (no separation between excitation and emission wavelengths) at a spectral resolution of 5 nm. Grams/AI 8.0 software (Thermo Electron Corporation; Waltham, MA, USA) was used to process the recorded data. The measurements were carried out at the Department of Biophysics, University of Life Sciences in Lublin. All spectral measurements were performed in triplicate at room temperature.

## 5. Conclusions

Among analyzed *1*,*3*,*4*-thiadiazole derivatives, C1 demonstrates the highest potential to increase the sensitivity of *Candida* cells to AmB. Compared with the other compounds, C1 shows better synergist interactions with AmB against selected *Candida* species. Notably, it enhances the antibiofilm activity of AmB. What is more, C1 also demonstrates better SI compared with NTBD, which indicates its greater efficacy against fungal cells than the toxicity against host cells. According to the detailed biomolecular and biophysical analysis, the mechanism underlying observed synergy is associated with disaggregation of AmB molecules induced by the *1*,*3*,*4*-thiadiazole derivatives, which elevates its efficacy in disrupting the cell wall integrity. However, in order to project the proposed combination into clinical use, further research into their in vivo efficacy and safety is necessary.

## Figures and Tables

**Figure 1 ijms-24-03430-f001:**
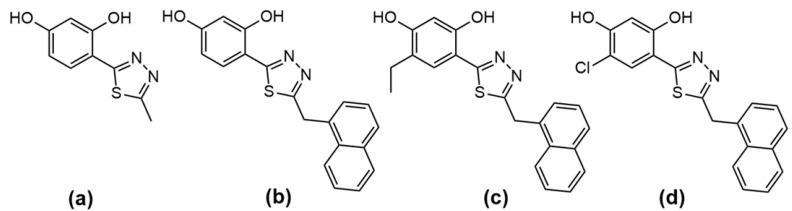
Structures of the investigated thiadiazole derivatives: C1 (**a**), NTBD (**b**); Et-NTBD (**c**), and Cl-NTBD (**d**).

**Figure 2 ijms-24-03430-f002:**
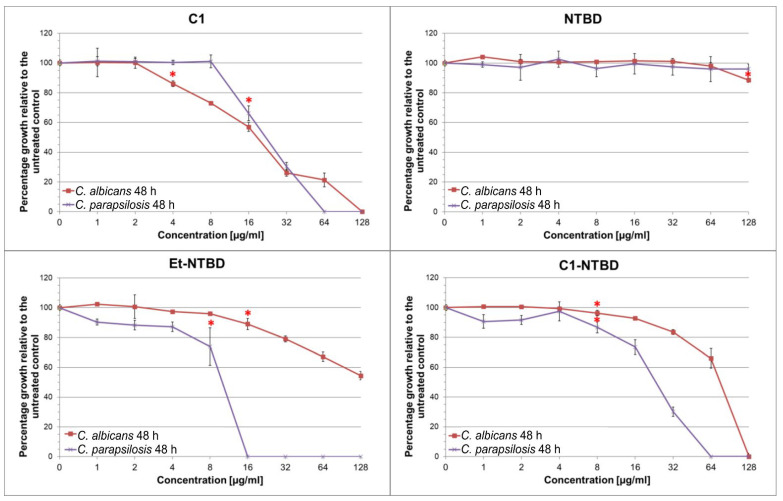
Effect of thiadiazoles on the growth of *C. albicans* NCPF 3153 and *C. parapsilosis* ATCC 22019. The percentage growth of thiadiazole-treated cells measured as OD600 relative to the control was calculated from the following equation: (OD600 of treated sample)/(OD600 of untreated sample) × 100; * represents the lowest concentration at which a statistically significant decrease in OD was noted, compared with the control (one-way ANOVA, Tukey’s post-hoc test).

**Figure 3 ijms-24-03430-f003:**
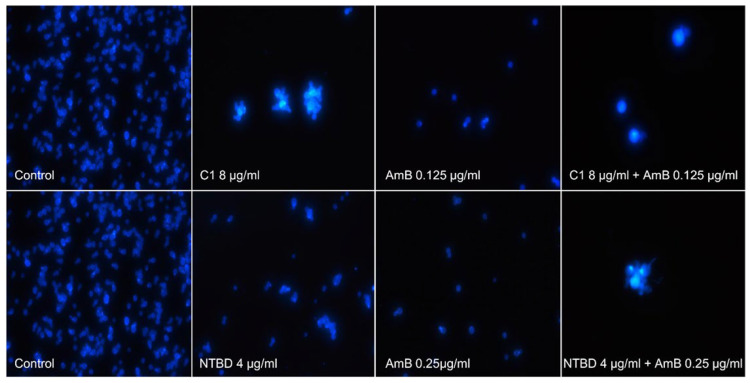
Fluorescence microscopy images showing morphological changes in *Candida* cells treated with C1 and NTBD separately and in combination with AmB at the selected concentrations. After 24 h of culture, the cells were stained with calcofluor white (magnification 600×).

**Figure 4 ijms-24-03430-f004:**
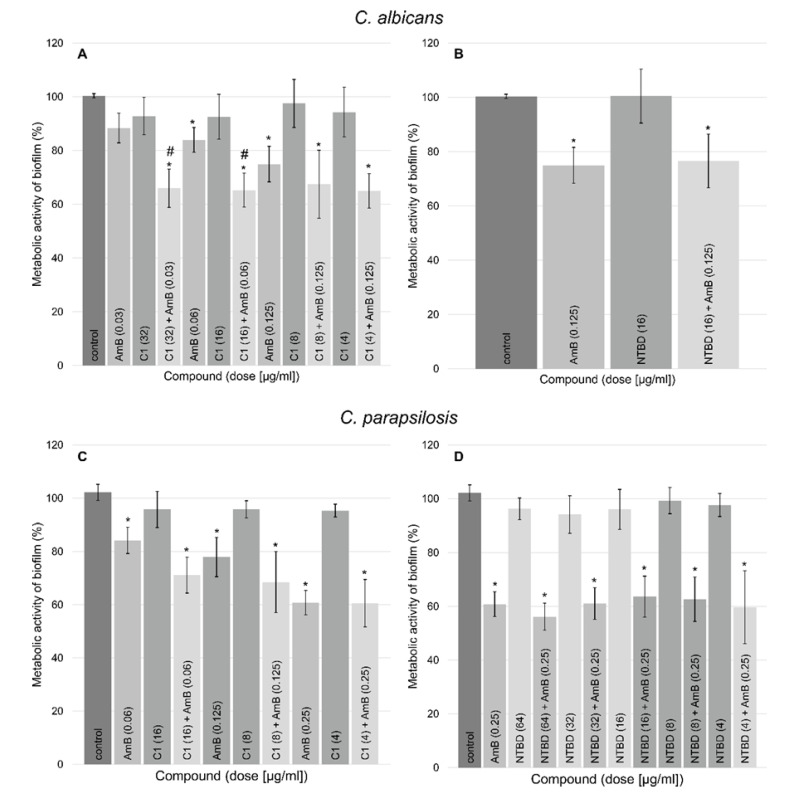
The MTT assay results presenting the analyzed compounds and combinations activity: (**A**)—AmB, C1, and C1+AmB activity against *C. albicans* biofilms, (**B**)—AmB, NTBD, and NTBD+AmB activity against *C. albicans* biofilms, (**C**)—AmB, C1, and C1+AmB activity against *C. parapsilosis* biofilms, (**D**)—AmB, NTBD, and NTBD+AmB activity against *C. parapsilosis* biofilms, *—the statistically significant differences (*p* < 5%) in the metabolic activity of *Candida* biofilms treated with AmB, thiadiazole derivatives and their combinations compared with the control, #—the statistically significant differences (*p* < 5%) in the metabolic activity *Candida* biofilms treated with thiadiazole derivatives + AmB compared with biofilms subjected to AmB.

**Figure 5 ijms-24-03430-f005:**
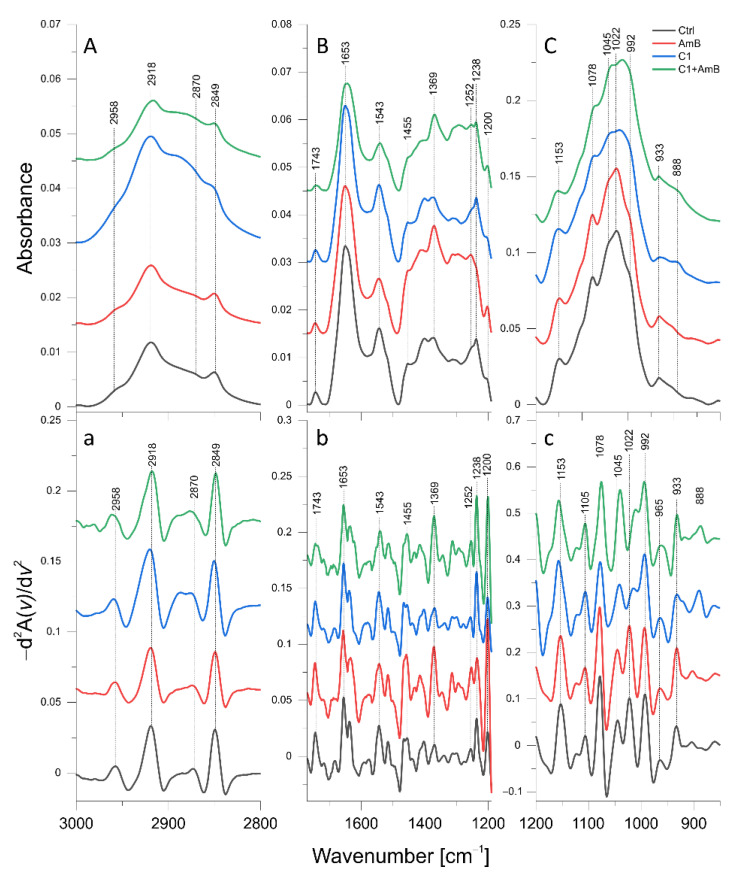
Mean ATR-FTIR spectra for the control *C. albicans* cells and cells treated with AmB (0.03 µg/mL), C1 (8 µg/mL), and their combination, respectively: (**A**)—3000–2800 cm^−1^ region of ATR-FTIR spectra; (**B**)—1750–1190 cm^−1^ region of ATR-FTIR spectra; (**C**)—1200–800 cm^−1^ region of ATR- FTIR spectra; (**a**–**c**)—reversed second derivatives of spectra presented in (**A**–**C**).

**Figure 6 ijms-24-03430-f006:**
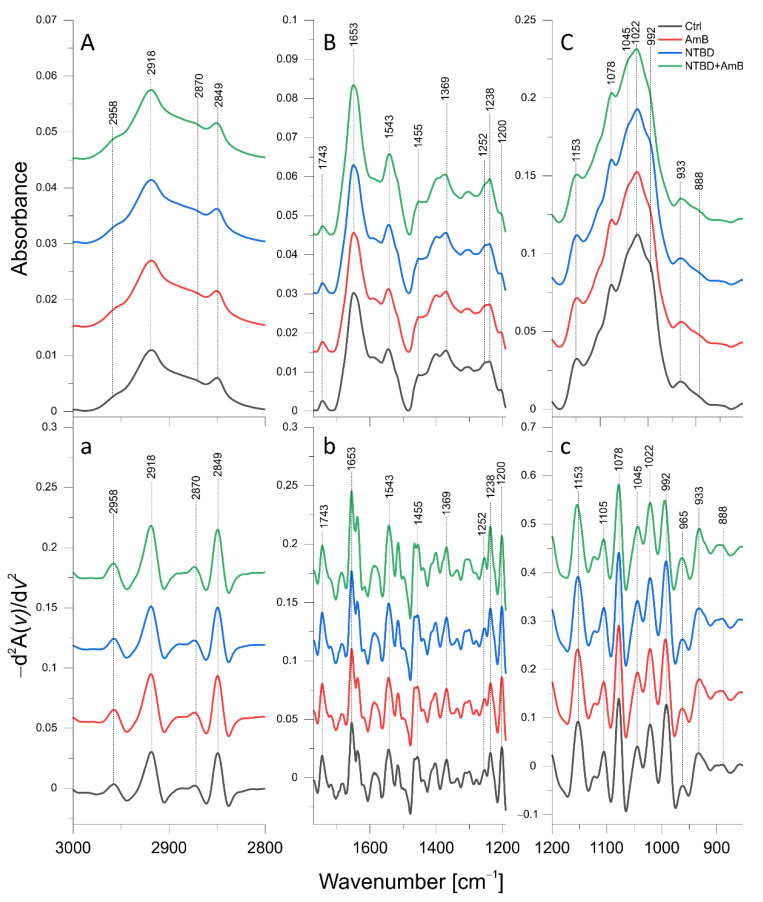
Mean ATR-FTIR spectra of control *C. albicans* cells and cells treated with AmB (0.06 µg/mL), NTBD (4 µg/mL), and their combination: (**A**)—3000–2800 cm^−1^ region of ATR-FTIR spectra; (**B**)—1750–1190 cm^−1^ region of ATR-FTIR spectra; (**C**)—1200–800 cm^−1^ region of ATR- FTIR spectra; a–c—reversed second derivatives of spectra presented in (**A**–**C**).

**Figure 9 ijms-24-03430-f009:**
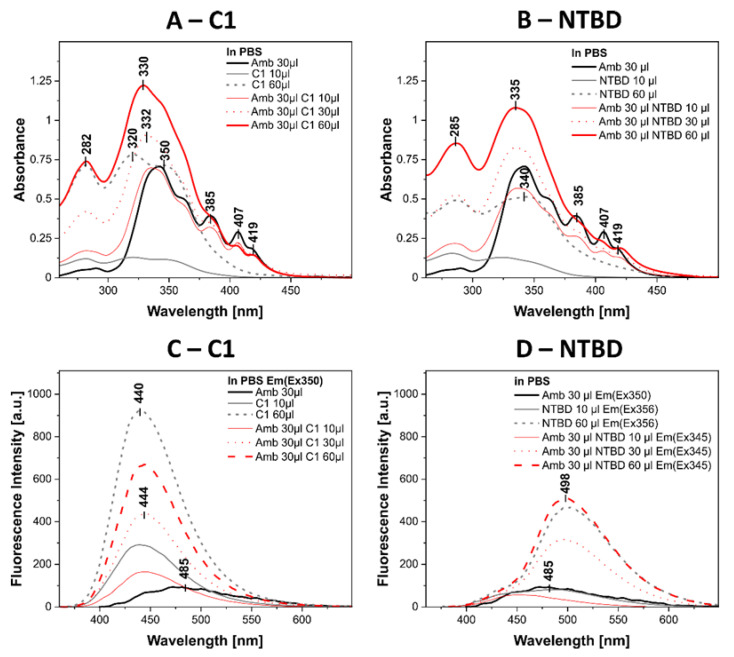
(**A**)—electron absorption spectra for C1, AmB, and the C1+AmB synergistic system in PBS buffer at varying concentrations of the compound. The molar concentration of AmB was 0.0107 mM, and the molar concentration of C1 was 0.0159 mM for the 10 μL dose and 0.0942 mM for 60 μL. The successive molar concentrations of C1 in the synergistic system were 0.0158 mM, 0.0471 mM, and 0.0933 mM; (**B**)—electron absorption spectra for NTBD, AmB, and the NTBD+AmB synergistic system in PBS buffer at varying concentrations of the compound. The molar concentration of AmB was 0.0107 mM, and the molar concentration of NTBD was 0.0099 mM for the 10 μL dose and 0.0587 mM for 60 μL. The successive molar concentrations of compound NTBD were 0.0098 mM, 0.0294 mM, and 0.0581 mM; (**C**)—electron emission spectra corresponding to the absorption spectra in A; (**D**)—electron emission spectra corresponding to the absorption spectra in B.

**Figure 10 ijms-24-03430-f010:**
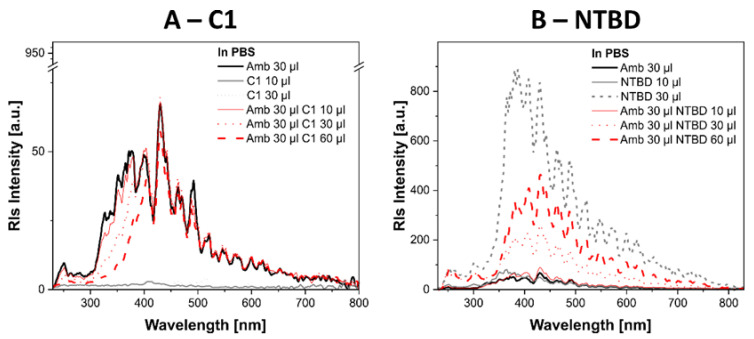
(**A**)—resonance light scattering spectra of C1, AmB, and the synergistic C1 + AmB system corresponding to the spectra in Figure 9A,C (analogous concentrations); (**B**)—resonance light scattering spectra of NTBD, AmB, and the synergistic NTBD + AmB system corresponding to the spectra in Figure 9B,D (analogous concentrations).

**Figure 11 ijms-24-03430-f011:**
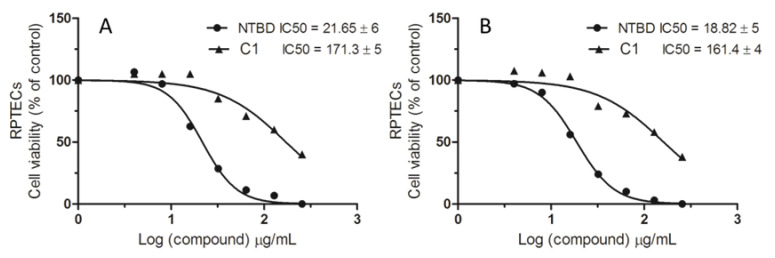
Fitting of drug dose-response curve to yield half-maximal inhibitory concentration. The cells were treated with C1 and NTBD (0–256 μg/mL) for 24 h (**A**) and 48 h (**B**) and MTT assay was performed. The IC50 value was estimated by non-linear regression analysis.

**Table 1 ijms-24-03430-t001:** Interactions between *1*,*3*,*4*-thiadiazoles (C1, NTBD, Et-NTBD, and C1-NTBD) and amphotericin B (AmB) in terms of activity against reference strains in vitro after 48-h incubation: MIC—minimal inhibiting concentration of the compound, ∑FIC—coefficient, which is the sum of partial concentrations inhibiting the growth of the pathogens. The interactions were determined based on the specified ranges: synergy (∑FIC ≤ 0.5), indifferent (∑FIC ≥ 0.5 to 4), antagonism (∑FIC > 4); ** = synergy.

Strain	*1*,*3*,*4*-thiadiazole	Separate Treatment		Combinatory Treatment *1*,*3*,*4*-thiadiazole +AmB											
				2 (µg/mL) of *1*,*3*,*4*-thiadiazole		4 (µg/mL) of *1*,*3*,*4*-thiadiazole		8 (µg/mL) of *1*,*3*,*4*-thiadiazole		16 (µg/mL) of *1*,*3*,*4*-thiadiazole		32 (µg/mL) of *1*,*3*,*4*-thiadiazole		64 (µg/mL) of *1*,*3*,*4*-thiadiazole	
		MIC *1*,*3*,*4*-thiadiazole (µg/mL)	MIC AmB (µg/mL)	MIC AmB (µg/mL)	ƩFIC	MIC AmB (µg/mL)	ƩFIC	MIC AmB (µg/mL)	ƩFIC	MIC AmB (µg/mL)	ƩFIC	MIC AmB (µg/mL)	ƩFIC	MIC AmB (µg/mL)	ƩFIC
** *C. albicans* ** **NCPF 3153**	C1	128	0.5	0.25	0.52	0.125	0.29 **	0.125	0.33 **	0.0625	0.29 **	0.0312	0.40 **	0.0312	0.73
	NTBD	>128	0.5	0.25	0.51	0.25	0.52	0.25	0.53	0.125	0.31 **	0.25	0.62 **	0.25	0.75
	Et-NTBD	>128	0.5	0.5	1.01	0.5	1.02	0.5	1.03	0.5	1.06	0.5	1.13	0.25	0.625
	C1-NTBD	128	0.5	0.5	1.01	0.5	1.03	0.5	1.06	0.5	1.13	0.5	1.25	0.25	1
** *C. parapsilosis* ** **ATCC 22019**	C1	64	1	0.5	0.53	0.25	0.31 **	0.125	0.25 **	0.0625	0.31 **	0.0312	0.53	-	-
	NTBD	>128	1	0.5	0.51	0.25	0.26 **	0.25	0.28 **	0.25	0.31 **	0.25	0.38 **	0.25	0.5 **
	Et-NTBD	16	1	1	1.125	0.5	0.75	0.5	1	-	-	-	-	-	-
	C1-NTBD	64	1	1	1.03	1	1.06	1	1.13	1	1.25	0.5	1	-	-

**Table 2 ijms-24-03430-t002:** Interactions of C1 with AmB against clinical strains in vitro after 48-h incubation. MIC—minimal inhibiting concentration of the compound, ∑FIC—coefficient, which is the sum of partial concentrations inhibiting the growth of the pathogens. The interactions were determined based on the specified ranges: synergy (∑FIC ≤ 0.5), indifferent (∑FIC ≥ 0.5 to 4), antagonism (∑FIC > 4); ** = synergy.

Strain	Separate Treatment		Combinatory Treatment C1 + AmB											
			2 (µg/mL) of C1		4 (µg/mL) of C1		8 (µg/mL) of C1		16 (µg/mL) of C1		32 (µg/mL) of C1		64 (µg/mL) of C1	
	MIC C1 (µg/mL)	MIC AmB (µg/mL)	MIC AmB (µg/mL)	ƩFIC	MIC AmB (µg/mL)	ƩFIC	MIC AmB (µg/mL)	ƩFIC	MIC AmB (µg/mL)	ƩFIC	MIC AmB (µg/mL)	ƩFIC	MIC AmB (µg/mL)	ƩFIC
***C. albicans* isolate 40**	>128	0.125	0.125	1.01	0.0625	0.52	0.0312	0.28 **	0.0625	0.56	0.0625	0.63	0.125	1.25
***C. parapsilosis* isolate 73**	128	0.25	0.0312	0.14 **	0.0312	0.16 **	0.0312	0.19 **	0.0312	0.25 **	0.0625	0.5 **	0.0625	0.75
** *C. krusei* ** **isolate 39**	>128	0.25	0.5	2.01	0.5	2.02	0.5	2.03	0.5	2.06	1	4.125	1	4.25
***C. glabrata* isolate 5**	>128	0.125	0.125	1.01	0.125	1.01	0.25	2.03	0.25	2.06	0.25	2.125	0.25	2.25

**Table 3 ijms-24-03430-t003:** Interactions of NTBD with AmB against clinical strains in vitro after 48-h incubation. MIC—minimal inhibiting concentration of the compound, ∑FIC—coefficient, which is the sum of partial concentrations inhibiting the growth of the pathogens. The interactions were determined based on the specified ranges: synergy (∑FIC ≤ 0.5), indifferent (∑FIC ≥ 0.5 to 4), antagonism (∑FI C> 4); ** = synergy.

Strain	Separate Treatment		Combinatory Treatment NTBD + AmB											
			2 (µg/mL) of NTBD		4 (µg/mL) of NTBD		8 (µg/mL) of NTBD		16 (µg/mL) of NTBD		32 (µg/mL) of NTBD		64 (µg/mL) of NTBD	
	MIC NTBD (µg/mL)	MIC AmB (µg/mL)	MIC AmB (µg/mL)	ƩFIC	MIC AmB (µg/mL)	ƩFIC	MIC AmB (µg/mL)	ƩFIC	MIC AmB (µg/mL)	ƩFIC	MIC AmB (µg/mL)	ƩFIC	MIC AmB (µg/mL)	ƩFIC
** *C. albicans* ** **isolate 40**	>128	0.125	0.125	1.01	0.125	1.02	0.125	1.03	0.125	1.06	0.125	1.125	0.125	1.25
***C. parapsilosis* isolate 73**	>128	0.25	0.25	1.01	0.25	1.02	0.25	1.03	0.25	1.06	0.25	1.25	0.5	2.25
** *C. krusei* ** **isolate 39**	>128	0.25	0.5	2.01	0.5	2.02	0.5	2.03	0.5	2.06	0.5	1.125	0.5	2.25
***C. glabrata* isolate 5**	>128	0.125	0.125	1.01	0.125	1.02	0.125	1.03	0.125	1.06	0.125	1.125	0.125	1.25

**Table 4 ijms-24-03430-t004:** IC50 and SI values of tested thiadiazoles (C1 and NTBD); *—MIC of thiadiazole derivatives alone; **—range of thiadiazole derivatives MIC values in the combination with AmB, which exhibited a synergistic effect with the antibiotic.

Drug	IC50 RPTECs (μg/mL)	Strain	Drug Concentration (μg/mL)	Selectivity Index (μg/mL)
C1	161.4 ± 4.0	*C. albicans*	128 *	1.26
			4–32 **	40.35–5.04
		*C. parasitosis*	64 *	2.51
			4–16 **	40.35–10.09
NTBD	18.82 ± 5.0	*C. albicans*	256 *	0.07
			16 **	1.18
		*C. parapsilosis*	256 *	0.07
			4–64 **	4.71–0.29

## Data Availability

Data reported in this manuscript will be available upon request.

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
