# Peer review of "Synergistic Antifungal Interactions between Antibiotic Amphotericin B and Selected 1,3,4-thiadiazole Derivatives, Determined by Microbiological, Cytochemical, and Molecular Spectroscopic Studies"

_ijms, 2023, doi:10.3390/ijms24043430_

Round 1
Reviewer 1 Report
Dear authors,
The article entitled: “Synergistic antifungal interactions of antibiotic amphotericin B with selected 1,3,4-thiadiazole derivatives determined by microbiological, cytochemical, and molecular spectroscopic studies” sounds great and actually it was well-designed. Since there is an increasing trend in emergence of resistant isolates as etiological agents of fungal diseases, performing these studies is needed. However, considering which antifungal should be evaluated with which other antifungal is an important issue. Here are my comments:
Why did you apply AmB as the reference drug to be combined with new derivatives? Why did you choose safer antifungals such as echinocandines?
Author Response
The article entitled: “Synergistic antifungal interactions of antibiotic amphotericin B with selected 1,3,4-thiadiazole derivatives determined by microbiological, cytochemical, and molecular spectroscopic studies” sounds great and actually it was well-designed. Since there is an increasing trend in emergence of resistant isolates as etiological agents of fungal diseases, performing these studies is needed. However, considering which antifungal should be evaluated with which other antifungal is an important issue. Here are my comments:
Why did you apply AmB as the reference drug to be combined with new derivatives? Why did you choose safer antifungals such as echinocandines?
Our experiments showed that the investigated thiadiazoles do not interact with echinocandins nor with azoles. Moreover, AmB was chosen because of its high efficacy and low rate of Candida strains resistance. AmB is toxic at higher concentrations but combining it with substances acting synergistically may allow to use lower concentrations of AmB, which will reduce its toxic effect and at the same time maintain high activity.

Reviewer 2 Report
In this manuscript author(s) has described the efficacy and potency of amphotericin B antibiotic combination with selected 1,3,4-thiadiazole residues. The authors made a thorough analysis of combinations of drugs with synergistic effect, and analyzed the mechanism of their combined effect on fungi. Combinations of C1 and NTBD derivatives with amphotericin B demonstrated valuable synergic antifungal interactions.
However, there are several minor drawbacks:
Line 64. "It has a broad spectrum of activity but is toxic at higher concentrations. AmB is mainly used in severe systemic mycoses of e.g. lungs and airways, the nervous system, and the gastrointestinal tract." It is better to specify what range of concentrations of AmB are demonstrating strong toxic effects, and what are not.
line 74. "Combination therapies are used to improve treatment outcomes and reduce drug toxicity; however, some studies have demonstrated antagonism between amphotericin B and azoles." Here it is worth clarifying what may be the reason for the greater efficacy of combination therapy and what molecular mechanisms underlie it, the same question is for antagonism.
Compound structures appears only in the end of the maniscript, but at the first time in the text we observe them in the introduction part. It would be more useful if the structures under investigation also would be demostrtated at the beginning of the article.
I consider the presented work valuable and after few corrections, I think that it could be published in the Journal.
Author Response
Reviewer 2
In this manuscript author(s) has described the efficacy and potency of amphotericin B antibiotic combination with selected 1,3,4-thiadiazole residues. The authors made a thorough analysis of combinations of drugs with synergistic effect, and analyzed the mechanism of their combined effect on fungi. Combinations of C1 and NTBD derivatives with amphotericin B demonstrated valuable synergic antifungal interactions.
However, there are several minor drawbacks:
Line 64. "It has a broad spectrum of activity but is toxic at higher concentrations. AmB is mainly used in severe systemic mycoses of e.g. lungs and airways, the nervous system, and the gastrointestinal tract." It is better to specify what range of concentrations of AmB are demonstrating strong toxic effects, and what are not. It has a broad spectrum of activity but is toxic at higher concentrations. AmB is mainly used in severe systemic mycoses of e.g. lungs and airways, the nervous system, and the gastroin-testinal tract.
The authors would like to thank the Reviewer for this valuable remarks. Indeed, AmB is a commonly used antifungal that has severe systemic toxicity. The safety infusion daily dose is limiting from 0.5 to 1.5 mg/kg/day depending of the pathogen, patient’s condition and severity of the infection. Higher antibiotics concentration can cause many side effects such as nausea, vomiting, cardiovascular disorders, and kidney damage 1. In vitro studies have shown that AmB concentrations ranging from 5 to 10 µg/mL caused abnormal morphology and reduced proliferation of osteoblasts and fibroblasts. Concentrations of 100 µg/mL and above caused cell death 2. The appropriate information was added in the introduction part of the manuscript using track-changes mode.
- Hamill, R.J. Amphotericin B formulations: a comparative review of efficacy and toxicity. Drugs 2013, 73, 919-934, doi:10.1007/s40265-013-0069-4.
- Harmsen, S.; McLaren, A.C.; Pauken, C.; McLemore, R. Amphotericin B Is Cytotoxic at Locally Delivered Concentrations. Clin Orthop Relat R 2011, 469, 3016-3021, doi:10.1007/s11999-011-1890-2.
line 74. "Combination therapies are used to improve treatment outcomes and reduce drug toxicity; however, some studies have demonstrated antagonism between amphotericin B and azoles." Here it is worth clarifying what may be the reason for the greater efficacy of combination therapy and what molecular mechanisms underlie it, the same question is for antagonism.
The main synergistic mechanisms may occur by increasing membrane permeability, reducing drug efflux from the cell, inhibiting biofilm formation, and/or inhibiting heat shock protein 90 (Hsp90) synthesis 3,4. On the other hand, drugs which inhibit ergosterol biosynthesis, such as fluconazole in combination therapy with AmB, may have an antagonistic effect due to the fact that ergosterol is a target molecule for AmB. This is a big problem in the treatment of mycoses because such drugs induce resistance of fungal pathogens to amB itself 5. The appropriate information was added in the introduction part of the manuscript using track-changes mode.
- Li X, Wu X, Gao Y and Hao L (2019) Synergistic Effects and Mechanisms of Combined Treatment With Harmine Hydrochloride and Azoles for Resistant Candida albicans. Front. Microbiol. 10:2295. doi: 10.3389/fmicb.2019.02295
- Kane, A.; Carter, D.A. Augmenting Azoles with Drug Synergy to Expand the Antifungal Toolbox. Pharmaceuticals 2022, 15, 482. https://doi.org/10.3390/ph15040482
- E. Lewis, D.J. Diekema, S.A. Messer, M.A. Pfaller, M.E. Klepser Comparison of Etest, chequerboard dilution and time–kill studies for the detection of synergy or antagonism between antifungal agents tested against Candida species J Antimicrob Chemother, 49 (2002), pp. 345-351
Compound structures appears only in the end of the maniscript, but at the first time in the text we observe them in the introduction part. It would be more useful if the structures under investigation also would be demostrtated at the beginning of the article.
According to the Reviewers suggestion, the structures of the thiadiazoles were moved to the beginning of the manuscript as Figure 1.
I consider the presented work valuable and after few corrections, I think that it could be published in the Journal.

Reviewer 3 Report
The article by Dróżdż et al. demonstrated the synergy between AmpB and 1,3,4-thiadiazole derivatives. Considering the present status of AMR especially in fungal strains, these studies are very important. Authors nicely addressed the issue of AMR in Candida spp., and demonstrated potent antifungal activity of thiadiazole derivatives in presence of AmpB. However, manuscript is many limitations and serious fundamental drawbacks in few experiments. Here are my overall comments.
Major comments
1. This study mainly focus on synergistic activities of thiadiazole derivatives and AmpB. However, the method use for the determination of synergy is very confusing. Please modified text and use following method to calculate the FICI. The interaction of test compound with antifungal agents referred to the fractional inhibitory concentration index (FICI). Arithmetically, ΣFIC's (Fractional inhibitory concentration) was calculated as: ΣFIC = FIC A + FIC B; where, FIC A is the MIC of drug A in the combination / MIC of drug A alone, and FIC B is the MIC of drug B in combination / MIC of drug B alone. The combination is considered synergistic if ΣFIC is ≤0.5, indifferent when the ΣFIC is ≥0.5 to 4, and antagonistic when the ΣFIC is >4 [Odds FC. Synergy, antagonism, and what the chequerboard puts between them. J Antimicrob Chemother (2003) 52:1.].
2. Synergy tables: the data presented is very misleading, Table should be organized as MIC of drug A alone, MIC of Drug B alone, Combination A and B, FICI indexes , FICI.
3. Please give the data for selectivity index in cytotoxicity study of these thiadazole derivatives.
3. Authors only demonstrated the in vitro MIC for these combinations, in order to project any of these combinations further, antibiofilm activity, in vivo efficacy is must. Please comment.
4. English has to be improved, please check whole manuscript with the native English speaker. Many typos, even with scientific words (Candidiasis, Mycosis), sentence fragments are quite visible in whole manuscript. Be consistent with the strains name, abbreviations. After explaining the Candida albicans once in the manuscript, use C. albicans, same for other strains. Amphotericin B, should be used as AmB.
Author Response
Reviewer 3
The article by Dróżdż et al. demonstrated the synergy between AmpB and 1,3,4-thiadiazole derivatives. Considering the present status of AMR especially in fungal strains, these studies are very important. Authors nicely addressed the issue of AMR in Candida spp., and demonstrated potent antifungal activity of thiadiazole derivatives in presence of AmpB. However, manuscript is many limitations and serious fundamental drawbacks in few experiments. Here are my overall comments.
Major comments
- This study mainly focus on synergistic activities of thiadiazole derivatives and AmpB. However, the method use for the determination of synergy is very confusing. Please modified text and use following method to calculate the FICI. The interaction of test compound with antifungal agents referred to the fractional inhibitory concentration index (FICI). Arithmetically, ΣFIC's (Fractional inhibitory concentration) was calculated as: ΣFIC = FIC A + FIC B; where, FIC A is the MIC of drug A in the combination / MIC of drug A alone, and FIC B is the MIC of drug B in combination / MIC of drug B alone. The combination is considered synergistic if ΣFIC is ≤0.5, indifferent when the ΣFIC is ≥0.5 to 4, and antagonistic when the ΣFIC is >4 [Odds FC. Synergy, antagonism, and what the chequerboard puts between them. J Antimicrob Chemother (2003) 52:1.].
The authors would like to thank the Reviewer for the valuable remarks. Analyzing the results of our experiments we referred to the studies of Isenberg 1992 and Karlowsky et al 2006 6,7. However, as the study indicated by the Reviewer undermines the existence of additivity effect, we modified the text following the Reviewers instructions. The changes were introduced in the manuscript using track-changes mode.
- Isenberg, H. D. E. Clinical Microbiology Procedures Handbook. American Society for Microbiology, Washington DC. (1992).
- Karlowsky, James A et al. “In vitro interactions of anidulafungin with azole antifungals, amphotericin B and 5-fluorocytosine against Candida species.” International journal of antimicrobial agents vol. 27,2 (2006).
- Synergy tables: the data presented is very misleading, Table should be organized as MIC of drug A alone, MIC of Drug B alone, Combination A and B, FICI indexes , FICI.
In the authors' opinion, modifying the tables in accordance with the reviewer's recommendations will reduce their readability. Because of the number of analysed combinations of thiadiazoles and AmB, organising the data applying the Reviewer’s instruction would force increase the number of presented tables without providing any additional, relevant information. However, some additional information, facilitating tables analysis were added in the their captions and body using track-changes mode. We hope that the Reviewer will accept the tables as they are presented.
- Please give the data for selectivity index in cytotoxicity study of these thiadazole derivatives.
The authors regret to inform that the data for selectivity index are not available upon presented study. Collecting such data would require repeating the experiment, which in turn would significantly extend the article proceeding. We hope that the article will be acceptable without this data. We thank the Reviewer for pointing out this issue and would like to assure you that data on selectivity index will be provided in our future articles.
- Authors only demonstrated the in vitro MIC for these combinations, in order to project any of these combinations further, antibiofilm activity, in vivo efficacy is must. Please comment.
The authors agree with Reviewers opinion that studies on antibiofilm activity and in vivo efficacy of the proposed combinations are necessary. Thus such experiments are planned to be done subsequently. The appropriate comment on this issue was added in the discussion part using track-changes mode.
- English has to be improved, please check whole manuscript with the native English speaker. Many typos, even with scientific words (Candidiasis, Mycosis), sentence fragments are quite visible in whole manuscript. Be consistent with the strains name, abbreviations. After explaining the Candida albicans once in the manuscript, use C. albicans, same for other strains. Amphotericin B, should be used as AmB.
According to the Reviewer remark, the extensive English correction was done using track-changes mode.

Round 2
Reviewer 3 Report
The authors answered many questions raised by all three reviewers. However, cytotoxicity data, and antibiofilm data both in vitro can be done which adds a lot to this manuscript. I understand the problem with in vivo efficacy for taking animal experiment approval and all.
Any compound that is cytotoxic no matter how good is at killing pathogens has no value to become a drug. So I would suggest the authors test the cytotoxicity of these compounds and calculate the selective index. The antibiofilm activity of these compounds in combination would also be a plus for the manuscript in order to project it as a future drug candidate.
At this moment I am rejecting the manuscript in its present form.
Author Response
Reviewer 3
The authors answered many questions raised by all three reviewers. However, cytotoxicity data, and antibiofilm data both in vitro can be done which adds a lot to this manuscript. I understand the problem with in vivo efficacy for taking animal experiment approval and all.
Any compound that is cytotoxic no matter how good is at killing pathogens has no value to become a drug. So I would suggest the authors test the cytotoxicity of these compounds and calculate the selective index. The antibiofilm activity of these compounds in combination would also be a plus for the manuscript in order to project it as a future drug candidate.
At this moment I am rejecting the manuscript in its present form.
According to the reviewer’s remarks the selectivity index of the tested compounds as well as their antibiofilm activity were determine and presented in the manuscript. The changes were introduced in the manuscript using track-changes mode. We sincerely hope that the completed manuscript will be accepted by the reviewer for publication.

Round 3
Reviewer 3 Report
The authors answered all my questions. It is evident from the SI that these compounds are cytotoxic to the mammalian cell lines. Especially, the compound NTBD has shown a SI of 0.07-4.71 which demonstrated selective toxicity for host cells compared to the pathogen. Generally, compounds showing SI of 10 or more are considered for the in vivo studies. However, C1 showed some promise to be considered for the in vivo studies as it showed better SI compared to NTBD. Please discuss this issue in the discussion section and project C1 as the lead compound. I would like to suggest the authors should write a conclusion of 4-5 lines at the end and project C1 as their main lead.
NTBD seems to be very toxic to human cell lines, I believe it would do the same in other cell lines as well including RBCs (Hemolysis).
Antibiofilm activity of C1 is encouraging. I am accepting the current MS after the minor changes as suggested.
Author Response
The authors answered all my questions. It is evident from the SI that these compounds are cytotoxic to the mammalian cell lines. Especially, the compound NTBD has shown a SI of 0.07-4.71 which demonstrated selective toxicity for host cells compared to the pathogen. Generally, compounds showing SI of 10 or more are considered for the in vivo studies. However, C1 showed some promise to be considered for the in vivo studies as it showed better SI compared to NTBD. Please discuss this issue in the discussion section and project C1 as the lead compound. I would like to suggest the authors should write a conclusion of 4-5 lines at the end and project C1 as their main lead.
NTBD seems to be very toxic to human cell lines, I believe it would do the same in other cell lines as well including RBCs (Hemolysis).
Antibiofilm activity of C1 is encouraging. I am accepting the current MS after the minor changes as suggested.
We would like to thank the Reviewer for the valuable suggestions. Changes to the manuscript have been made in accordance with the reviewer's comments in tracking change mode. We hope that now manuscript will fulfil the requirements for publication in IJMS.
